# Assessing the Impact of a Shinrin-Yoku (Forest Bathing) Intervention on Physician/Healthcare Professional Burnout: A Randomized, Controlled Trial

**DOI:** 10.3390/ijerph192114505

**Published:** 2022-11-04

**Authors:** John Kavanaugh, Mark E. Hardison, Heidi Honegger Rogers, Crystal White, Jessica Gross

**Affiliations:** 1Department of Psychiatry and Behavioral Sciences, University of New Mexico Hospital, University of New Mexico School of Medicine, Albuquerque, NM 87106, USA; 2Occupational Therapy Graduate Program, University of New Mexico, Albuquerque, NM 87131, USA; 3College of Nursing, University of New Mexico College of Nursing, Albuquerque, NM 87131, USA; 4Clinical and Translational Science Center, University of New Mexico Health Science Center, Albuquerque, NM 87131, USA

**Keywords:** forest bathing, Shinrin-Yoku, forest therapy, professional burnout, healthcare providers

## Abstract

Professional healthcare worker burnout is a crisis in the United States healthcare system. This crisis can be viewed at any level, from the national to local communities, but ultimately, must be understood at the level of the individual who is caring for patients. Thus, interventions to reduce burnout symptoms must prioritize the mental health of these individuals by alleviating some of the symptoms of depression, grief, and anxiety that accompany burnout. The practice of Shinrin-Yoku (Forest Bathing) is a specific evidence-based practice which research has shown can improve an individual’s mental health and, when performed in a group, can support a sense of social connection. We investigated the impact of a three-hour, guided Shinrin-Yoku (Forest Bathing) nature-based intervention on burnout symptoms among physicians and other healthcare workers by using a randomized, controlled trial. The Oldenburg Burnout Inventory (OLBI) and Mini-Z assessments were used to collect baseline burnout scores and participants were randomized into the intervention group, which completed the assessment again after the Shinrin-Yoku walk, or into a control group, which completed the assessments again after a day off from any clinical duties. A total of 34 participants were enrolled in the intervention group and a total of 22 participants were enrolled in the control group. Ultimately, no statistically significant differences were detected between the pre-test and post-test scores for the intervention group or between the post-test scores of the intervention group compared to the control group. However, the subjective responses collected from participants after participating in the Shinrin-Yoku walk overwhelmingly reported decreased feelings of stress and increased mental wellbeing. This raises important questions about the difference between symptoms of burnout and other aspects of mental health, as well as the limitations of a one-time nature-based intervention on levels of chronic burnout symptoms. Thus, further research on the effects of engaging healthcare providers in an ongoing practice of Shinrin-Yoku is warranted.

## 1. Introduction

Shinrin-Yoku (forest bathing) is the practice of “visiting a forest or engaging in various therapeutic activities in a forest environment to improve one’s health and wellbeing” [1]. More specifically, it is an intentional practice in which the practitioner immerses themselves in nature while mindfully giving attention to the sensory information they receive from that natural environment. Forest Bathing as a term was coined by the Japanese government in 1982, and since this time, researchers around the world have been assessing the impact of Forest Bathing on a wide variety of physiological and psychological variables. These include potential benefits for immune system function (increase in natural killer cells/cancer prevention), the cardiovascular system (hypertension/coronary artery disease), the respiratory system (allergies and respiratory disease), depression and anxiety (mood disorders and stress), mental relaxation (attention deficit/hyperactivity disorder) and human feelings of “awe” (increase in gratitude and selflessness) [2]. In a study conducted in Japan in 2019 to evaluate the effects of Forest Bathing on working age adults with high stress and depressive tendencies, a two-hour Forest Bathing activity was shown to decrease blood pressure and alleviate negative psychological parameters, especially in participants with depressive tendencies [3]. Some benefits from a single Forest Bathing experience, such as the improved functioning of natural killer (NK) cells, have been found to persist for up to 30 days [4]. Based on the demonstrated health benefits of Forest Bathing, researchers have stated the need for further research in the “areas of healthcare professional stress reduction and life balance” [2].

Professional burnout is of increasing concern within the healthcare system, both in terms of the mental health of the providers as well as the impact it has on the quality and safety of patient care. Burnout is an unfortunately prevalent syndrome among physicians and other healthcare workers and the symptoms can consist of emotional exhaustion, depersonalization, and feelings of low personal accomplishment [5]. It is estimated that 33–50% of all physicians experience at least one of these dimensions of burnout and that these symptoms are also prevalent among other healthcare workers [5,6]. This is especially relevant given that work-related stress and symptoms of burnout have been exacerbated by the COVID-19 pandemic [7]. Multiple studies and organizations have mobilized to understand and address the issue of physician burnout. In a 2019 study on the effectiveness of resident-led initiatives, residents reported that they perceived the most effective interventions to be fitness-based activities (gym, walk, outdoor activity, Tai Chi) and art therapy. In a meta-analysis of different interventions, a meditation workshop was also noted to decrease resident burnout rates [8,9]. These findings are highly suggestive that Forest Bathing, which involves outdoor activity with a mindful meditative component, could be beneficial in addressing physician burnout.

In a review of the literature, only one study was identified which analyzed the effects of a Forest Therapy intervention on burnout. A 2015 study held in Korea studied the effects of a 3-day Forest Therapy program on 19 female “workers in the healthcare and counseling service industry”, who reported a low-frequency use of a forest environment compared to a control group of 20 female participants from the same industry who reported a high-frequency use of a forest environment [10]. The control group was initially found to have lower indicators of stress than the experimental group, which was thought to be secondary to them already using the forest environment more frequently prior to the study. The findings of the study were variable; however, it did find an improvement in perceived stress levels and on professional efficacy in participants of the Forest Therapy camp as determined by the Maslach Burnout Inventory (MBI). This study and the lack of similar studies demonstrates the need for further research of this therapy as a potential intervention to address the serious and widespread prevalence of physician burnout.

In the United States, the Association of Nature and Forest Therapy (ANFT) is a leading organization providing Forest Therapy interventions. It offers courses for participants to become certified Forest Therapy guides, who then lead others through a series of invitations to encourage a therapeutic, mindful engagement with the natural environment [11]. These guided walks introduce new practitioners to Forest Therapy who are unfamiliar with this therapeutic exercise. This has similarities to the Forest Therapy centers in countries such as Japan, which provide guides, on-site physicians, and designated Forest Bathing trails with signs listing specific invitations into the practice along the route.

The purpose of this study was to evaluate the effect of a guided Forest Bathing session on symptoms of burnout and collect qualitative feedback on this intervention. It was hypothesized that a Forest Bathing session would result in a statistically significant improvement from baseline burnout symptoms when compared to a control group experiencing a typical day off of clinical work.

## 2. Materials and Methods

### 2.1. Design

This project was a two-arm, randomized, controlled trial with a waitlist control group. Outcomes were collected at two timepoints: before and after the allocated intervention. Participants were allocated into intervention groups by using the randomization module on REDCap that draws from a pre-randomized table, assigning sequential participant IDs to different groups. A blinded researcher allocated all new participants at the time they entered the study by clicking the “randomize” button on the REDCap electronic form. The randomization table this process drew on was generated before the study by using blocks of 6 for all possible permutations for 2 groups, and then rolling dice to generate the sequence of blocks.

The study was conducted according to the guidelines of the Declaration of Helsinki and approved by the Institutional Review Board of the University of New Mexico Health Sciences Center (21-419, approved on 1 February 2022. Informed consent was obtained from all subjects involved in the study before beginning. All recruitment, interventions, and data collection occurred in a 3-month period in Spring 2022. Participants in the experimental group were allowed to sign up for a Forest Bathing walk immediately after filling out the pre-test materials. Participants in the waitlist control group had to wait a minimum of 5 days and fill out the post-test survey on a day off from clinical duties before signing up for a Forest Bathing walk. This did not move them into the experimental group, but rather was carried out solely to allow all interested individuals to participate in the intervention. This option was made available given that this intervention was being offered to assist with the wellness of healthcare providers coping with the challenges of a worldwide pandemic.

### 2.2. Participants and Recruitment

Recruitment used convenience sampling of health sciences faculty and medical residents at a university/teaching hospital in the southwestern United States. Possible participants were contacted through email announcements and in-person presentations on the physiological and psychological benefits of nature immersion/Forest Bathing. To be included in the study, participants needed to be: (1) at least 18 years of age, (2) current faculty of the university’s Health Sciences Center or working as a medical resident at the teaching hospital, (3) able to physically tolerate a 0.8 km (0.5 mile) walk.

### 2.3. Interventions

The Forest Bathing walks were conducted through a collaboration with two Forest Bathing guides who had completed the certification course through the Association of Nature and Forest Therapy (ANFT). They were hired to be the interveners for the project. The Forest Bathing walks were initially held at a public open space encompassing approximately 525 hectares (1300 acres) of mountains and a piñon-juniper forest located 30 min from the university hospital (see Figure 1). The elevation of the intervention area is approximately 2285 m (7500 feet). The walk began in an open meadow of grama grass (*Bouteloua* species) speckled with paintbrush flowers (*Castillega* species) and overlooked by several large ponderosa pines (*Pinus ponderosa*). It then wandered through scattered groves of one-seed junipers (*Juniperus monosperma*), pinyon pines (*Pinus edulis*), and Gambel oak (*Quercus gambelli*). The walk concluded at a dramatic rise populated by multiple alligator junipers (*Juniperus deppeana*) growing among large boulders, which provided a view of the forest-covered Sandia mountains. Due to extensive wildfires, public lands were closed during the period of this study and the final walks were invited to be held on nearby private land with similar vegetation and terrain. The course of the walk in both locations was less than 0.8 km (0.5 miles) on a dirt path with some mild variations in grade. Participants were sent an email with information on the walk as well as suggestions for weather preparedness (temperatures ranged from approximately 10 degrees Celsius (50 degrees Fahrenheit) in the mornings of the first walks to 27 degrees Celsius (80 degrees Fahrenheit) in the afternoons of the last walks) and safety. In addition, a short PowerPoint presentation on the practice of Forest Bathing was provided. The ANFT guides started the experience with introductions, followed by an orientation and safety talk, to help all participants feel comfortable with the intervention. The participants were guided in a 15 min grounding and centering exercise in order to help set the stage for the following “invitations” which were offered, and in order to guide intentional awareness of the senses as the participants moved slowly along the path. These invitations followed the standard sequence of an ANFT-based walk and included noticing and interacting with their surroundings. The participants were brought together in a circle after each invitation and were given the opportunity to share what they were noticing. This practice serves to reinforce and deepen the connection of the social group, as well as model and inspire further connections with the natural surroundings. The ANFT sequence provides some standardization of the intervention while also allowing the guide to adapt the specific therapeutic practices to the participants and environment. The final invitation was for a 20 min “sit spot”, where participants had the opportunity to sit and notice their natural surroundings. The experience concluded with a gratitude practice and shared tea. At the end of the intervention, participants discussed the experience together and asked questions of the ANFT guide before walking back together. The entire experience was 3 h. Groups of Forest Bathing walkers were mixed such that the ANFT guide was blinded to which participants were walking as a part of the experimental group or the control group.

### 2.4. Measures

Data collected for the study included three elements: (1) demographic information; (2) two standardized assessments of burnout; (3) open-ended questions soliciting feedback about the intervention.

The Oldenburg Burnout Inventory (OLBI) assessment is a standardized assessment used to assess the level of burnout in physicians providing direct patient care. The OLBI is a widely used measure of burnout because it uses 16 positively and negatively formulated items to measure exhaustion and disengagement [12]. These items further demonstrate that burnout can be interpreted in terms of the “identification continuum”, which provides a dimension ranging from disengagement to dedication, as well as the “energy continuum”, which provides a dimension ranging from exhaustion to vigor. The OLBI has been proven effective for both work and academic settings, and for all employees, not just those in healthcare [13]. It has been found to have a high reliability (Cronbach’s alpha = 0.63), especially concerning the two dimensions it evaluates: exhaustion (Cronbach’s alpha = 0.87) and disengagement (Cronbach’s alpha = 0.81) [12].

The Mini-Z is a questionnaire on work-related burnout symptoms, modified from the MEMO (Minimizing Error Maximizing Outcome) questionnaire, where prior work-life measures are assessed [14]. The Mini-Z questionnaire includes 10 items that address job satisfaction, in which participants indicate potential burnout predictors. The Mini-Z has been evaluated for reliability and validity through the annual administration to all departments at Hennepin County Medical Center in Minneapolis, MN [15]. Reliability is reasonable, with Cronbach’s alpha of 0.8 when participants answer using a 1–5 scale within the range from strongly disagree to strongly agree, respectively (e.g., “I feel a great deal of stress because of my job”) [16]. Satisfaction and stress are good predictor variables that correlate with overall burnout (*p* < 0.001) [17].

### 2.5. Analysis

Summary statistics were calculated for interventions and controls and compared prior to analysis using the Wilcoxon rank-sum and Fisher’s exact tests to confirm the groups were matched demographically.

We analyzed OLBI scores and Mini-Z scores for all individuals in the sample and standardized the reversed scores on the OLBI questionnaire prior to analysis so that the scale was consistent for all questions. Scores for the two questionnaires were tested for normality and homogeneity of variances using the Shapiro–Wilk and Bartlett’s tests, respectively.

After confirming the data normality and homogeneity of variances, we conducted Student’s *t*-tests to determine whether baseline and post-test scores differed within the intervention and control groups, and Welch’s *t*-tests to determine whether baseline scores for interventions and controls differed from each other and whether the intervention led to significantly different post-test scores for the two study groups. We analyzed both the total scores for each questionnaire, as well as the OLBI disengagement and exhaustion subdomains. *p*-values were adjusted for multiplicity using the Benjamini–Hochberg method.

All data were collected and compiled in REDCap before being exported for analyses in R version 4.1.1.

## 3. Results

A total of 34 participants were enrolled in the intervention group and a total of 22 participants were enrolled in the control group. During the study, several participants were lost to follow-up, and as a result, 10 people from the intervention group and 12 people from the control group were excluded. Participants in the intervention and control groups were matched for age, gender, ethnicity, and race. See Table 1 for complete demographic data.

The baseline burnout scores of the control and intervention groups from the pre-test are similar and show no statistical difference between the two groups. Scores from the OLBI range from 16 to 64. Past studies have a score of 35 or above as a positive indicator of burnout [18]. The median scores for the pre-tests and post-tests of both the intervention group and the control group were all above 35 on the OLBI, which is consistent with the presence of burnout symptoms in these groups. The Mini-Z evaluates burnout as well as satisfaction and stress, and a total score of 40 or above is defined as a joyful workplace [19]. Neither the pre-test nor post-test median scores for either group reached this threshold score of 40. Question 2 of the Mini-Z asks the participant to use their own definition of burnout to choose from five options, and the results can be seen in Figure 2.

When analyzing the data from the OLBI, we did not find a statistically significant difference between the intervention group and the control group or between the pre-test and post-test scores for the intervention group (see Table 2). When analyzing the data from the Mini-Z assessment, there was a slight but statistically significant decrease in the post-test scores of the intervention group relative to the control group. However, after adjusting the *p*-values, this finding was no longer significant. The Mini-Z results similarly did not demonstrate a significant difference between the pre- and post-test results in either the intervention or the control group. There was also no significant decrease in post-test scores for the intervention group relative to the control group for the burnout symptom subdomains of exhaustion and disengagement on the OLBI (see Table 3).

## 4. Discussion

The aim of this study was to assess the impact of participating in a Shinrin-Yoku intervention on the level of burnout of physicians and other healthcare workers. Findings that would have clearly demonstrated that this therapy can decrease burnout symptoms would have shown a significant decrease in burnout scores on the post-test relative to the pre-test in the intervention group on either the OLBI or the Mini-Z tests, or a significant decrease in the burnout scores on the post-test for the intervention group relative to the control group. The latter was initially detected for the Mini-Z test alone, although this finding was then negated when the *p*-value was adjusted. This result may, in part, be due to a loss of power in the context of a smaller sample size. Although the results from the Mini-Z test are suggestive that the intervention decreased aspects of burnout, it would require further studies with a larger sample size to better determine this. Overall, the results from this study do not demonstrate a significant impact on burnout scores from these two tools, resulting from a Forest Bathing intervention.

The data add to the body of evidence that burnout symptoms are prevalent among healthcare providers. The median scores from the OLBI are consistent with burnout symptoms in both the intervention group and the control group. On question 2 of the Mini-Z test, 34% of participants reported some symptoms of burnout. This includes 12% who reported persistent burnout symptoms that do not go away.

The post-intervention data also gathered subjective comments on the mental state of the Shinrin-Yoku participants. Twenty participants responded with comments. Nineteen of these comments focused on feeling more relaxed, more peaceful, calmer, more appreciative, and less anxious. Only one comment stated that they did not detect a difference in their internal state after the walk.

These results that do not demonstrate a significant difference in burnout scores after a Shinrin-Yoku intervention appear to be in contradiction to these subjective comments which report the psychologically calming benefits of this therapy. These findings also appear in contradiction to the results from other studies that showed participating in Shinrin-Yoku can alleviate symptoms of depression, anxiety, and stress, which are all connected to symptoms of burnout [2]. This then raises the question of the nature of healthcare worker burnout and how it is fundamentally different from either a state of mood or a psychiatric disorder. It can be described as a reactive syndrome that develops over a prolonged period due to systemic issues, bureaucratic burdens, and repetitive moral injury. One of the participants of this study put this succinctly when they wrote, “I felt very relaxed after forest therapy. Then I came home and did 6 h of work that needed to be done before Monday, and then took the survey. I generally feel stressed on Sunday afternoons trying to finish all the things that didn’t get done during the week and forest therapy didn’t take those things away.” It is likely that, for some, their experience of a Shinrin-Yoku walk highlighted rather than relieved their sources of burnout symptoms. It is also of note that recruitment was a significant challenge for this study, not due to a lack of interest, but rather due to difficulties for many interested physicians in finding sufficient time off from their clinical duties to join a Shinrin-Yoku walk. This represented an insurmountable obstacle for many of the 79 people who originally signed up as interested in the study. As one participant wrote after their Shinrin-Yoku walk, they were “more aware of how exhausted I am, the degree of anxiety I carry around. I also felt more at peace and like I might be able to tolerate the intense 120+ hour schedule I have for the next 2 weeks”. This articulates both the challenge in addressing the source of burnout symptoms and the need for interventions such as Shinrin-Yoku for providing the subjective sense of peace that is so often lacking in healthcare. It is clear from the subjective responses that many of the participants on these Shinrin-Yoku walks found them to be of value to their mental wellbeing despite not significantly impacting their burnout symptoms.

When considering why the study participants reported feelings of “peace” that did not translate to decreased burnout scores, it is important to note that Shinrin-Yoku is often proposed to be an ongoing practice rather than a one-time intervention [20]. The distinguished Japanese researchers of this therapy, Dr Yoshifumi Miyazaki and Dr Qing Li, as well as Amos Clifford, the founder of the Association of Nature and Forest Therapy, propose that Shinrin-Yoku be incorporated into the participant’s life as an ongoing practice [11,20]. In a way similar to psychotherapy, yoga, and tai chi, it is understandable that a single session of Shinrin-Yoku would not produce a measurable improvement in a chronic condition, such as healthcare worker burnout. This, as well as the small sample size, were significant limitations for this study which introduced this therapy to dozens of healthcare workers, but did not assess the long-term impact of using this therapy over months and years on burnout symptoms.

## 5. Conclusions

In conclusion, the data from this randomized controlled trial did not demonstrate a change in burnout symptoms from participating in a single Shinrin-Yoku walk when compared to baseline burnout scores or when compared to a control group. This does not mean, however, that there is no possible role for Forest Therapy in providing relief from burnout symptoms. The subjective comments from participants after the Shinrin-Yoku walk are consistent with many previous studies which demonstrated the ability of this practice to decrease stress and anxiety [2]. It is unlikely that anyone can experience burnout without simultaneously experiencing stress and anxiety. Burnout symptoms accumulate over time, and thus, it is reasonable to suppose that alleviating burnout will also take time. An ongoing practice of Forest Therapy may very well erode the severity of burnout symptoms through cumulative improvements in stress and anxiety levels. Therefore, there is a need for further research on the impact of Shinrin-Yoku on burnout symptoms in healthcare workers over a prolonged period and as a regular practice.

## Figures and Tables

**Figure 1 ijerph-19-14505-f001:**
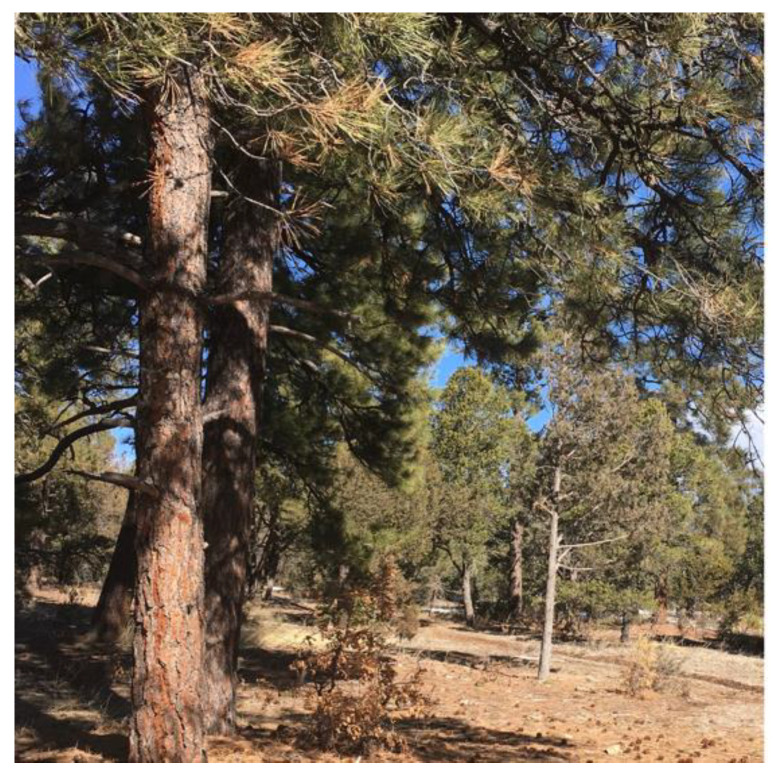
The Forest Bathing walks took place primarily in ponderosa pine, pinyon, and juniper habitat, as can be seen in this picture of the intervention site.

**Figure 2 ijerph-19-14505-f002:**
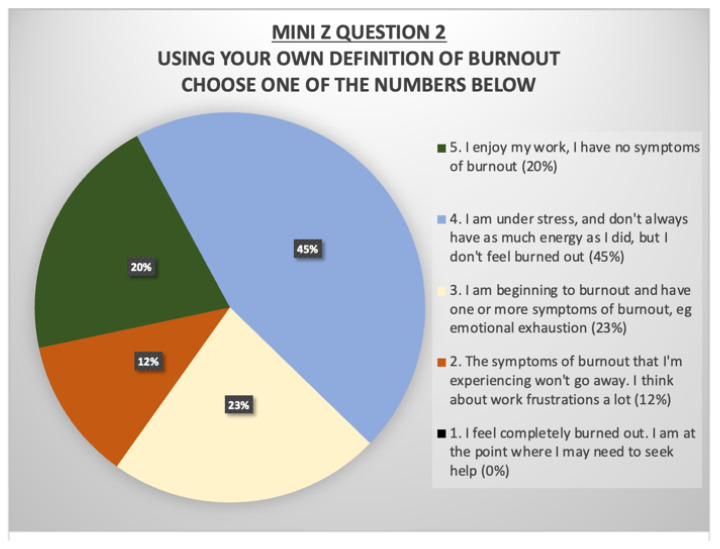
Results from all participants on Mini-Z Question 2 concerning burnout symptoms.

**Table 1 ijerph-19-14505-t001:** Demographic information.

		Cases (*n* = 34)	Controls (*n* = 22)	*p*	Test for Group Comparison
Age in Years: Mean (Standard Deviation)		40.50 (12.7)	38.73 (8.0)	0.81	Wilcoxon Rank-Sum
		*n* (%)	*n* (%)		
Gender	Nonbinary	0 (0.0%)	1 (4.6%)	0.26	Fisher’s exact
	Female	27 (79.4%)	19 (86.4%)		
	Male	7 (20.6%)	2 (9.1%)		
	None of these	0 (0.0%)	0 (0.0%)		
Hispanic/Latino	Yes	2 (5.9%)	4 (18.2%)	0.20	Fisher’s exact
	No	32 (94.1%)	18 (81.8%)		
Race	Asian	5 (14.7%)	2 (9.1%)	0.56	Fisher’s exact
	Black or Af Am	2 (5.9%)	0 (0.0%)		
	Nat Am	2 (5.9%)	0 (0.0%)		
	Nat Hawaiian/PI	0 (0.0%)	0 (0.0%)		
	White	23 (67.7%)	19 (86.4%)		
	More than one race	0 (0.0%)	0 (0.0%)		
	Prefer not to answer	2 (5.9%)	1 (4.6%)		
Notes: Participant 55 does not have a group number, so is excluded from the table; participant 24 (case group) chose Race 3 and Race 5)					
Age breakdown					
	Full Sample	Cases	Controls		
Mean Age	39.8	40.5	38.7		
Minimum Age	26	26	28		
Maximum Age	69	69	65		
Sample Size by Age Range	*n* (%)	*n* (%)	*n* (%)		
Total	57 (100.0%)	35 (100.0%)	22 (100.0%)		
Age 20–29	8 (14.0%)	7 (20.6%)	1 (4.6%)		
Age 30–39	23 (40.4%)	12 (35.3%)	10 (45.5%)		
Age 40–49	18 (31.6%)	8 (23.5%)	10 (45.5%)		
Age 50–49	3 (5.3%)	3 (8.8%)	0 (0.00%)		
Age 60–69	5 (8.8%)	4 (11.8%)	1 (4.6%)		
Participant 65 is missing age (case group)					

**Table 2 ijerph-19-14505-t002:** Pre- and post-test analysis of Oldenburg Burnout Inventory and Mini-Z assessments.

**Oldenburg Burnout Inventory Overall Scores (Cases vs. Controls)**					
	**Cases**	**Controls**	**Test**	***p*-Value**	**adj *p*-Value**
pre-Test Median score	39.00	42.00	Wilcoxon rank-sum test	0.07	ns
post-test median score	38.00	41.00	Wilcoxon rank-sum test	0.17	ns
Notes: Reversed scores were recoded prior to analysis.					
**Mini-Z Scores (Cases vs. Controls)**					
	**Cases**	**Controls**	**Test**	***p*-Value**	**adj *p*-Value**
pre-test median score	35.00	36.00	Wilcoxon rank-sum test	0.21	ns
post-test median score	34.00	37.00	Wilcoxon rank-sum test	0.03	ns
Significant results are highlighted in red (raw *p* = 0.05 was used as the threshold for significance).					
**Oldenburg Burnout Inventory Overall Scores (Change in Scores within Groups)**					
	**Pre-Test Median Score**	**Post-Test Median Score**	**Test**	***p*-Value**	**adj *p*-Value**
cases (*n* = 24)			paired *t*-test		
controls (*n* = 10)			paired *t*-test		
cases (*n* = 24)	39.50	38.00	Wilcoxon signed-rank test	0.30	ns
controls (*n* = 10)	41.50	41.00	Wilcoxon signed-rank test	0.44	ns
Notes: Ten cases and twelve controls had pre-test scores, but not post-test scores; one case had a post-test score, but no pre-test score; these participants were excluded from the paired analyses. Reversed scores were recoded prior to analysis.					
**Mini-Z Scores (Change in Scores within Groups)**					
	**Pre-Test Median Score**	**Post-Test Median Score**	**Test**	***p*-Value**	**adj *p*-Value**
cases (*n* = 24)	33.63	33.92	paired *t*-test	0.43	0.43
controls (*n* = 10)	36.10	36.90	paired *t*-test	0.27	0.27
cases (*n* = 24)	34.00	34.00	Wilcoxon signed-rank test	0.23	ns
controls (*n* = 10)	37.00	37.00	Wilcoxon signed-rank test	0.31	ns
Notes: Ten cases and twelve controls had pre-test scores, but not post-test scores; one case had a post-test score, but no pre-test score; these participants were excluded from the paired analyses.					

**Table 3 ijerph-19-14505-t003:** Analysis of Oldenburg Burnout Inventory Subdomains.

**Oldenburg Burnout Inventory Subscores (Cases vs. Controls)**					
	**Cases**	**Controls**	**Test**	***p*-Value**	**adj *p*-Value**
Disengagement					
pre-test median score	18.00	20.00	Wilcoxon rank-sum test	0.07	ns
post-test median score	18.00	20.00	Wilcoxon rank-sum test	0.30	ns
Exhaustion					
pre-test median score	20.50	22.00	Wilcoxon rank-sum test	0.10	ns
post-test median score	20.00	21.50	Wilcoxon rank-sum test	0.04	ns
Notes: Reversed scores were recoded prior to analysis. Significant results are highlighted in red (raw *p* = 0.05 was used as the threshold for significance).					
**Oldenburg Burnout Inventory Subscores (Change in Scores within Groups)**					
	**Pre-Test Median Score**	**Post-Test Median Score**	**Test**	***p*-Value**	**adj *p*-Value**
Disengagement					
cases (*n* = 24)	19.00	18.50	Wilcoxon rank-sum test	0.33	ns
controls (*n* = 10)	20.00	20.00	Wilcoxon rank-sum test	0.88	ns
Exhaustion					
cases (*n* = 24)	20.00	20.00	Wilcoxon signed-rank test	0.34	ns
controls (*n* = 10)	21.50	21.50	Wilcoxon signed-rank test	0.16	ns
Notes: Ten cases and twelve controls had pre-test scores, but not post-test scores; one case had a post-test score, but no pre-test score; these participants were excluded from the paired analyses. Reversed scores were recoded prior to analysis.					

## Data Availability

Raw data can be obtained by contacting the primary author. All data will be de-identified to protect participant privacy.

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
