# Peer review of "Assessing the Impact of a Shinrin-Yoku (Forest Bathing) Intervention on Physician/Healthcare Professional Burnout: A Randomized, Controlled Trial"

_ijerph, 2022, doi:10.3390/ijerph192114505_

Round 1

Reviewer 1 Report

Paper is well-written but I'm concerned about the sample size and did not find the conclusions very convincing.

Author Response

Thank you very much for reviewing our submission. I have noted that the sample size is a limitation in line 335 and the conclusion has been rewritten.

Reviewer 2 Report

The topics – how to address the prevalent burnout in healthcare and the impact and efficacy of forest bathing on mental health, general health, and well-being – are both timely and relevant, their combination as a research area is a great idea. The study design is okay, with both pre- and post-intervention measures and comparison, and a control group. The results are presented comprehensibly.

I have a few requirements though to cover some current shortcomings.

-        Affiliations (numbers) are not indicated at the names of the authors.

-        2 and 3 are the same.

-        Crystal White does not appear in the Author Contributions.

-        Study design: I didn’t really understand why participants in the control group signed up for an FB walk after filling out the post-test survey (line 121). Was it just an opportunity to take part in the experience even if they had been randomized in the control group?

-        Interventions: ‘primarily pine and juniper’ is not sufficient information to characterize the vegetation of an FB route, since there are studies presenting varying effects of different forests. Please list the specific species that mainly compose the characteristic vegetation.

-        Please provide the altitude of the area where the FB walks were accomplished.

-        Provide at least the mean temperature in spring in that region, or the specific temperatures of the FB walks, or the general temperature in the FB walks time interval this year.

The above are necessary to make this study’s intervention comparable to others.

-        In line 174, which figure does “(figure 1)” refer to? Figure 1 is a photo of the intervention site. Is it in the appendix? Then state it related to this other figure 1.

-        Results: no data on the age of the participants are provided, neither numbers for gender, or ethnicity, or race. Only a significant difference is described between the two groups regarding gender. It’s natural that some people must be excluded from a human study during the process, which often results in the inhomogeneity of the intervention and the control groups. Fortunately, the pre-test scores were statistically similar between them, which makes them acceptable. Nevertheless, demographic data must be provided in such studies. Please include a small table containing number or type of genders, age, ethnicity, and race within the intervention group (n=24), as well as the control group (n=10).

-        I think there’s no need for the title “3.2. Figures, Tables and Schemes” in line 247. (On the one hand, there’s no 3.1., on the other hand Tables 1-2, and Figure 2. are integrated in 3. Results section, moreover, Figure 1. is elsewhere.)

-        Discussion: I suggest merging the “4. Discussion” section and “5. Conclusions” section as one “4. Discussion” section, and write a new, short Conclusions section. Otherwise, I like the discussion. It’s not a surprise that a single session of FB walk could not resolve burnout – I think chronic conditions require chronic (regular) ‘treatment’/intervention, but FB is a good direction. I also agree that it’s worth examining the individual

(subjective) reactions beside the statistics since individual variations/attitudes should also be concerned. The theory that (healthcare worker) burnout might have a special, complex nature is interesting; FB may help, but maybe it cannot be resolved without changing the basic, underlying, everyday working conditions.

-        Conclusions: please write a much shorter synthetizing section with the major points: no significant pre-post and intervention vs control findings, but favourable subjective responses; complexity and uniqueness of healthcare worker burnout; potential need for a regular practice; further research.

After completion of the presentation of the study and results and conclusions the work is worth being published in IJERPH, and it especially fits the special issue New Advances on Wellness Therapies Using Integrated Health Focusing on Nature, and it would be of interest both to researchers and to the public.

Author Response

Hello, thank you very much for taking the time to review our submission and for your excellent suggestions. 

  • I have corrected and clarified author contributions and affiliations
  • I clarified the study design in lines 123-126
  • I added the altitude, temperatures, and predominant plant species to the interventions section to help make this intervention comparable to other studies
  • "figure 1" originally in line 174 has been removed
  • demographic information has been added as table 1
  • the title “3.2. Figures, Tables and Schemes” has been removed
  • the original conclusion section was merged with the discussion section and I wrote a shorter conclusion section as suggested

Reviewer 3 Report

From the subjective responses that the authors investigated in this study, Sinlin-Yoku is presenting results such as stress reduction, but this result is generally suggested by studies already conducted. The authors will also want to use qualitative research methods to present subjective responses as conclusions.

At the same time, the results of the Oldenburg Burnout Inventory and Mini-Z show no statistically significant difference.
The authors need to present more results to support their conclusions based on evidence.

Author Response

Thank you for taking the time to review our submission. We agree that the results of our study do not support our hypothesis that a single forest bathing walk would significantly impact burnout scores. However, we find that there is merit in publishing negative results which in the case of this study can help grow our understanding of the nature of burnout and the limitations of a one-time intervention. I have rewritten the conclusions.

Round 2

Reviewer 2 Report

Dear Responder,

The manuscript has improved a lot, and now the presentation of the study is really professional (although I use °C and m but of course F and feet are also objective and international). :)

I accept it in present form.

Your work is a good initiative of healthcare burnout, and a great advance for the effects of forest walking.

Author Response

Thank you so much! I have now converted everything to metric units and will re-submit. Have a great day!

Reviewer 3 Report

Thank you for your efforts.

You used "Shinrin-yoku" for the title of the paper.
I don't think this term has gained enough sociality yet.
More citations are needed to use this term. At the same time, comparisons and reviews with similar terms are needed.
If this process is not possible, I recommend that you use terms or general terms that are gaining more sociality.

Author Response

Thank you for taking the time to review our article. After performing a pubmed search there were 47 articles including Shinrin-Yoku in the title. I have updated the document to use the hyphenated form as this is more consistent with existing research. An internet search found Shinrin-Yoku written about in Time, the Washington Post, and the Guardian. We feel this demonstrates a growing cultural awareness of this term and the practice it describes. We add to this by defining the practice in the first sentence of our introduction and by describing in detail the practice used in our Interventions section. Our study specifically uses Shinrin-Yoku as the intervention and, we hope, will continue to raise awareness about this practice.